METHODS AND RESOURCES

# Thinking small: Next-generation sensor networks close the size gap in vertebrate biologging

**Simon P. Ripperger**[1,2,3]*, **Gerald G. Carter**[2,3], **Rachel A. Page** *Supervision*[2], **Niklas Duda**[4], **Alexander Koelpin**[5], **Robert Weigel**[4], **Markus Hartmann**[6], **Thorsten Nowak**[6], **Jörn Thielecke**[6], **Michael Schadhauser**[6], **Jörg Robert**[6], **Sebastian Herbst**[7], **Klaus Meyer-Wegener**[7], **Peter Wägemann**[8], **Wolfgang Schröder-Preikschat**[8], **Björn Cassens**[9], **Rüdiger Kapitza**[9], **Falko Dressler**[10], **Frieder Mayer**[1,11]

1 Museum für Naturkunde–Leibniz Institute for Evolution and Biodiversity Science, Berlin, Germany, 2 Smithsonian Tropical Research Institute, Ancón, Republic of Panama, 3 Department of Evolution, Ecology, and Organismal Biology, The Ohio State University, Columbus, Ohio, United States of America, 4 Institute for Electronics Engineering, Friedrich-Alexander University Erlangen-Nürnberg (FAU), Erlangen, Germany, 5 Chair for Electronics and Sensor Systems, Brandenburg University of Technology, Cottbus, Germany, 6 Institute of Information Technology (Communication Electronics) LIKE, Friedrich-Alexander University Erlangen-Nürnberg (FAU), Erlangen-Tennenlohe, Germany, 7 Department of Computer Science, Friedrich-Alexander University Erlangen-Nürnberg (FAU), Erlangen, Germany, 8 Department of Computer Science, Friedrich-Alexander University Erlangen-Nürnberg (FAU), Erlangen, Germany, 9 Carl-Friedrich-Gauß-Fakultät, Technische Universität Braunschweig, Braunschweig, Germany, 10 Heinz Nixdorf Institute and Dept. of Computer Science, Paderborn University, Paderborn, Germany, 11 Berlin-Brandenburg Institute of Advanced Biodiversity Research, Berlin, Germany

* simon.ripperger@gmail.com

**Data Availability Statement:** All relevant data, code, custom software, and description of hardware are within the paper, its Supporting Information files, or have been archived by the

## Abstract

Recent advances in animal tracking technology have ushered in a new era in biologging. However, the considerable size of many sophisticated biologging devices restricts their application to larger animals, whereas older techniques often still represent the state-of-the-art for studying small vertebrates. In industrial applications, low-power wireless sensor networks (WSNs) fulfill requirements similar to those needed to monitor animal behavior at high resolution and at low tag mass. We developed a wireless biologging network (WBN), which enables simultaneous direct proximity sensing, high-resolution tracking, and long-range remote data download at tag masses of 1 to 2 g. Deployments to study wild bats created social networks and flight trajectories of unprecedented quality. Our developments highlight the vast capabilities of WBNs and their potential to close an important gap in biologging: fully automated tracking and proximity sensing of small animals, even in closed habitats, at high spatial and temporal resolution.

## Introduction

Recent advances in animal tracking technology have ushered in a new era in biologging [1,2]. By collecting data of unprecedented quantity and quality, automated methods have revolutionized numerous fields, including animal ecology [3], collective behavior [4], migration [5], and conservation biology [6]. For example, automated tracking of animals from space has advanced considerably over the past decade, in particular for observing large-scale movements

German Federation for Biological Data (GFBio). The following references give access to our GFBio depositions. Ripperger, S.; Nowak, T.; Hartmann, M.; Schadhauser, M. (2019). Bat Biologging Data. [Dataset]. Data Publisher: Museum für Naturkunde Berlin (MfN) - Leibniz Institute for Evolution and Biodiversity Science. https://doi.org/10.7479/vd6t-7a92. Ripperger (2019). Meeting Splitter [Software]. Data Publisher: Museum für Naturkunde Berlin (MfN) - Leibniz Institute for Evolution and Biodiversity Science. https://doi.org/10.7479/ytdf-wf05. Wägemann, P.; Ripperger, S. (2020). Ground node design for BATS tracking system. [Software]. Data Publisher: Museum für Naturkunde Berlin (MfN) - Leibniz Institute for Evolution and Biodiversity Science. https://doi.org/10.7479/z5ym-kx58.

**Funding:** This study was funded by grants of the Deutsche Forschungsgemeinschaft (FM, AK, RW, RK, KMW, WSP, JT, JR, FD; https://www.dfg.de/) within the research unit FOR-1508, a Smithsonian Scholarly Studies Awards grant (RAP, GGC, SPR, FM; https://www.si.edu/), and a National Geographic Society Research Grant WW-057R-17 (GGC; https://www.nationalgeographic.com/). The funders had no role in study design, data collection and analysis, decision to publish, or preparation of the manuscript.

**Competing interests:** The authors declare that no competing interests exist.

**Abbreviations:** AoA, angle of arrival; ARTS, automated radio-telemetry system; ECG, electrocardiogram; GPS, global positioning system; GSM, Global System for Mobile Communications; ICARUS, international cooperation for animal research using space; MAC, media access control; SNR, signal to noise ratio; WBN, wireless biologging network; VHF, very high frequency; WSN, wireless sensor network.

[1]. However, satellite communication for localization or data access requires a lot of power, and heavy transmitters greatly limit the ability to track smaller vertebrate species [1]. Efforts to further miniaturize increasingly powerful biologging devices culminated in the launch of the international cooperation for animal research using space (ICARUS) initiative, which aims to achieve global animal observation at a small tag mass through a combination of global positioning system (GPS) tracking, on-board sensing, energy harvesting, and energy-efficient data access from low space orbit [7]. ICARUS promises a great step forward in tracking large-scale movements such as migration. GPS tracking, however, is often not ideal or feasible for field biologists studying behavior on smaller spatial scales. GPS tracking of small vertebrate species is further limited by the considerable mass of GPS devices [1]. Satellite reception is hampered by complex habitats and impossible if animals go inside trees, caves, or underground burrows.

In industrial applications or for civilian surveillance, low-power wireless sensor networks (WSNs) fulfill requirements similar to those needed to track animal behavior at high resolution and at low tag mass [8]. Consistently, there have been numerous applications for WSNs in wildlife monitoring ("biologging") since the early 2000s [9]. In the last decade, more sophisticated approaches have created powerful monitoring systems, e.g., for high-resolution tracking [10] and fully automated logging of social encounters [11,12]. The major challenge in developing efficient wireless biologging networks (WBNs) is to design ultra-low power communication networks in order to maximize performance, minimize energy consumption, and reduce tag mass.

Here, we describe a multifunctional and modular system that takes WBNs to the next level (Fig 1). We first present a solution for direct proximity sensing that enables the collection of proximity data at a temporal resolution of seconds, at tag masses of 1 to 2 g, and with runtimes of up to several weeks (depending on the sampling rate). Second, we describe an adaptive option for triangulating spatial positions based on received signal strength by ground-borne localization nodes. This adaptive option allows automated recording of robust movement trajectories even in structurally complex habitats. Third, we explore a new, almost energy neutral solution for remote data access over distances of several kilometers at low data rates. Finally, we present an energy model that shows the effect of the parameter settings of software tasks on the runtime of the animal-borne tag. First deployments of this wireless biologging system have resulted in proximity and tracking data of unprecedented quality and have demonstrated the high potential of WBNs for studying (social) behavior. Our developments highlight the vast capabilities of WBNs and their potential to close an important gap in biologging: fully automated tracking and proximity sensing of small animals, even in closed habitats, at high spatial and temporal resolution.

## Results

The modular structure of the described system allows researchers to combine proximity sensing, long-range telemetry, and high-resolution tracking (Fig 1) depending on the research question and the behavior of the animals. We chose bats to test and validate the system because they are small-bodied and move fast in dense vegetation, both challenges to the performance of the WBN. Three recent field studies were conducted in temperate and tropical habitats on 3 bat species: greater mouse-eared bats (*Myotis myotis)*, common noctule bats (*Nyctalus noctula*), and common vampire bats (*Desmodus rotundus*). Each study documented high-resolution proximity data by direct proximity sensing among animals and automatically forwarding data to ground nodes (Fig 1A) that were deployed at roosting or foraging sites. Bat-borne mobile nodes that came within the reception range of the localization grid automatically increased their sampling rate to enable high-resolution localization (Fig 1B). Data for the

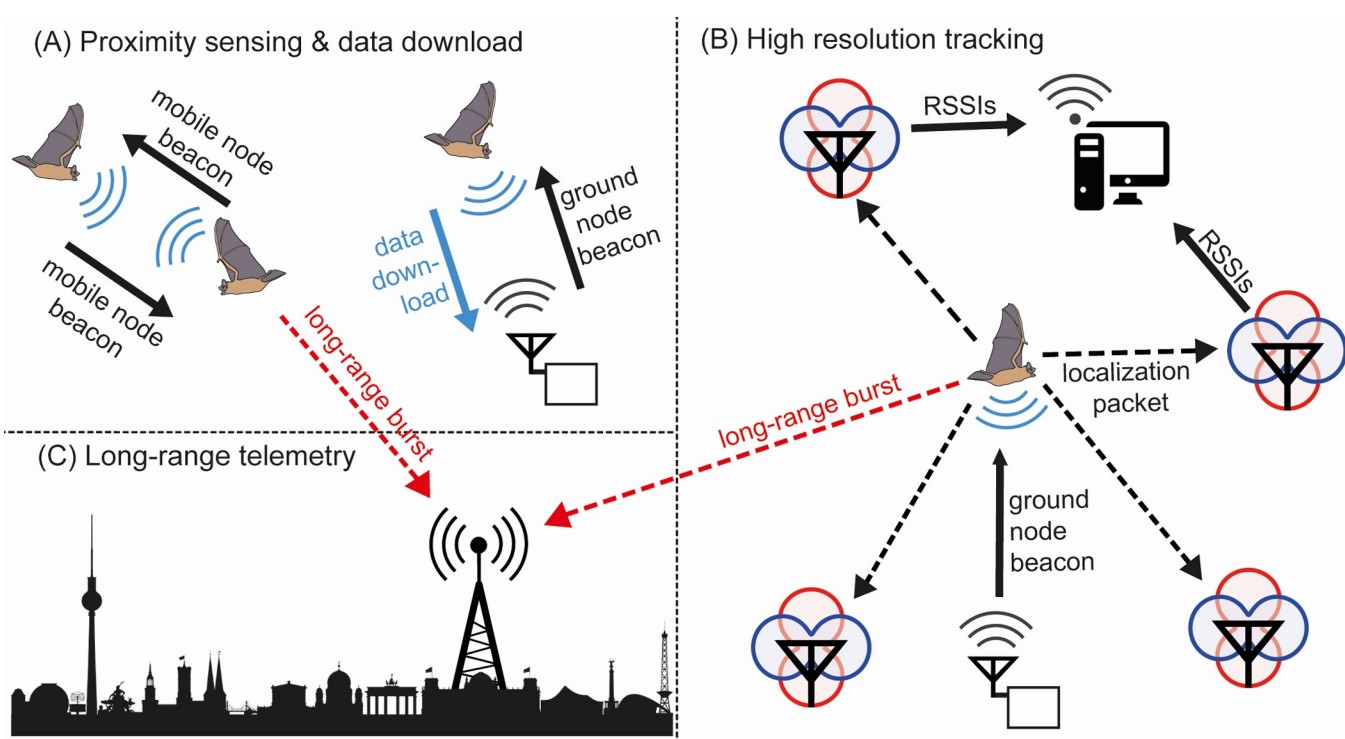

**Fig 1. Wireless biologging system overview.** (A) Animal-borne mobile nodes document animal-animal meetings, which are triggered by mobile-node beacons 24/7 and independently of the ground infrastructure. Each mobile node forwards its meeting data when it receives beacons from a ground node that is dedicated to downloading and storing. (B) When a tagged animal enters a grid of localization nodes (depicted by an antenna with red/blue gain patterns), a beacon of a tracking-dedicated ground node triggers the transmission of localization packets from the mobile node to the localization nodes. RSSIs of the impinging localization packets are then sent from the localization nodes to a work station via a WLAN. (C) Long-range bursts, which contain encoded sensor data, are received by long-range receivers. Long-range telemetry enables data transmission over distances of several kilometers at a low data rate. RSSI, received signal strength indicator; WLAN, wireless local area network.

synchronization of clocks was transmitted successfully over distances of more than 4 km by long-range telemetry (Fig 1C). The following sections describe the empirical validation of the system.

## High-resolution social network data from direct proximity sensing

Fifty individuals of one large natural colony of common vampire bats (*D. rotundus*) were tagged simultaneously in Panama. Associations with other tagged bats are fluid and highly dynamic both during day and night. For example, Fig 2A shows the course of the meeting history and the dynamic range of degree centrality for a single bat (ID 56) over a 2-day period. The high temporal resolution of meetings (all mobile nodes in reach communicate with each other every 2 s) also makes it possible to infer a behavior such as departure from the roost or movement within the roost. For example, foraging bouts can be identified by a sudden drop in meeting partners at night, which can be verified by contacts to ground nodes outside the roost. Autonomous direct proximity sensing allows monitoring changes in roosting associations, caused by moving among subgroups within the roost (Fig 2B and 2C), and it also allows inferring "social foraging networks" outside the roost (Fig 2D). In addition, every meeting is labeled with a maximum signal strength intensity indicator. This makes it possible to subset the meeting data set according to signal strength, an estimate for proximity [13]. Accordingly, one can distinguish close-contact associations from associations based on merely occupying the same area [14].

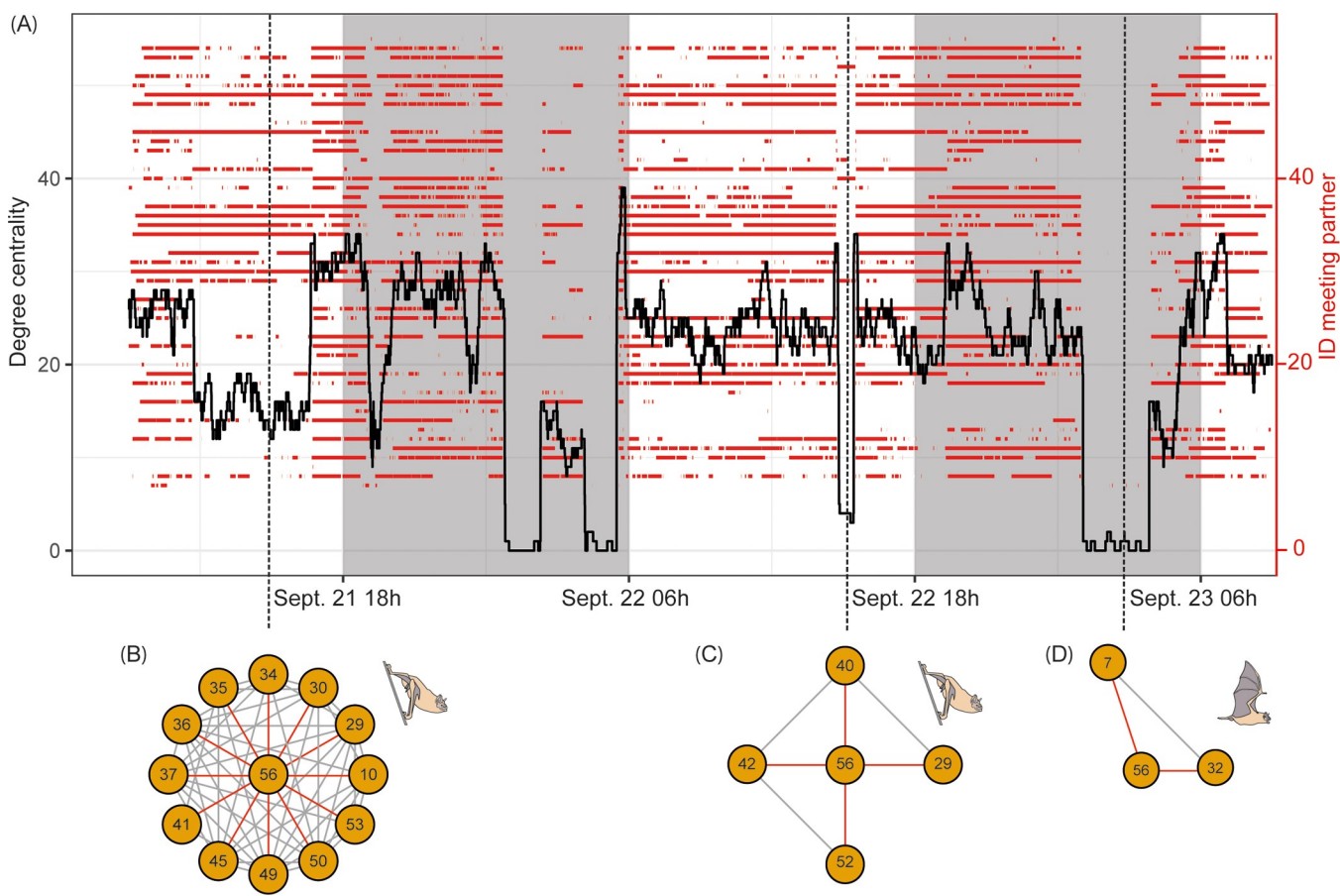

**Fig 2. High-resolution association data in wild vampire bats.** (A) Meeting history of a single vampire bat (ID 56; 50 tagged bats in total) with other tagged bats. Red lines show meetings between bat 56 and other tagged bats (right-hand y-axis). The black line shows the degree centrality (number of associated tagged bats, left-hand y-axis) of bat 56 every 2 s. Date and time are on the x-axis. Shaded areas indicate night time. Vertical dashed lines show egocentric social networks at each snapshot of time during roosting (B, C) and foraging (D). Associations with the focal bat are indicated by red lines. Data and software used to create this figure have been archived by GFBio (https://doi.org/10.7479/vd6t-7a92; https://doi.org/10.7479/ytdf-wf05).

The social networks created from direct proximity sensing are independent of the whereabouts of the tagged bats and provide an adaptive temporal resolution of seconds. Almost 400,000 individual meetings were recorded during the first 8 days of our field test. The typical approach for collecting social network data from bats has been to observe some or all of the bats within each roost in a sample of identified roosts each day; however, these co-roosting associations (e.g., the works by Wilkinson and colleagues and Wilkinson [15,16]) cannot detect social structure for individuals sharing the same single roost throughout a study. Our system, however, allows complete networks of spatial proximity of all bats every few seconds. This temporal resolution makes changes in social gatherings directly visible if time slices in high-resolution data are small enough [17]. We believe this represents an extraordinary advance for studying such small free-ranging animals, and it allows for an analytical depth that is so far known predominantly from human social networks generated by communication among smart phones or social media [17]. As an example, we have been able to gain a deeper understanding of the intrinsic and extrinsic factors influencing the stability of social relationships in vampire bats by combining captive experiments and proximity sensing in the wild [14]. Tracking associations at high temporal resolution in the wild allowed us to gather

evidence for rare and cryptic events, such as maternal guidance in noctule bats [18] and unfamiliar kin recognition in vampire bats [19].

## Received signal strength–based localization from angle-of-arrival estimation

Seventeen wireless localization nodes were used to track tagged 11 mouse-eared bats (*M. myotis*) over an area of approximately 1.5 ha in an old, natural deciduous forest in northern Bavaria (Germany, Forchheim). We were able to reconstruct flight trajectories from foraging mouse-eared bats. Fig 3B shows as an example 2 trajectories of one foraging mouse-eared bat during 2 different nights in early August.

We evaluated the spatial resolution of the tracking system by estimating a trajectory from a defined reference path using unscented Kalman filters. The reference path and estimated trajectory are shown in Fig 3A. The trajectories were calculated from angle-of-arrival estimates of signals impinging on localization nodes. Angles were estimated from difference measurements of received signal strength at 2 orthogonal antenna gain patterns. This procedure in combination with a set of postprocessing techniques for probabilistic multipath mitigation makes the trajectories robust to multipath propagation. The calculated trajectory is based on 4,912 data sets, and 1 set was composed of up to $2 \times 17$ received signal strength difference measurements (1 per frequency band) if all 17 localization nodes were within the reception range of the mobile node. For comparison, we also analyzed 4 tracks recorded by a 15 g heavy-duty Ornitela GPS tracker, which is commonly used for tracking large birds of up to 450 g body mass. The mean positioning error was 7.30 m for the Ornitela GPS tracker and 5.65 m for the trajectory of the WBN.

We calculated the positioning accuracy at lower densities of the localization grid. Localization was less accurate with fewer localization nodes (Fig 4), but it was robust and comparable to the full tracking grid (17 nodes) when using 15 to 16 nodes. With 12 to 14 nodes, we observed increasing variation in average error rates. With 11 nodes, the mean error was similar to the results from GPS tracking. Variation increased steadily with lower numbers of nodes, and the mean error reached more than 10 m with a maximum error rate of 34 m at 6 nodes. At such low grid densities, the localization results tended to diverge, resulting in increasing positioning errors. In addition, sparser grids lack robustness against multipath scattering. Consequently, the node density may only be reduced to a certain point, although positioning errors remain quite stable (Fig 4).

These analyses show that 11 localization nodes over an area of 1.5 ha in a forested habitat might be sufficient to construct high-resolution trajectories comparable in quality to a heavy-duty GPS tracker, which would only last for a few hours using a 15 g device, or to reverse GPS in open desert habitats [10]. Only moderate resources and human effort are needed to cover an area of a few hectares. For example, a setup as described above consisting of 11 localization nodes is deployed and configured by 2 people in 1 to 2 workdays.

## Distance verification of packet transmission by long-range telemetry

Transmission distances of data packages were measured in an urban green area. Thirty-four noctule bats (*N. noctula*) were tagged in a forest within the city of Berlin, and 2 long-range telemetry receivers were placed at a distance of about 1 and 4 km from the forest.

Mobile nodes transmitted a burst for long-range transmission of the mobile node's time stamp coupled with every mobile node beacon. During a period of 2 weeks, we were able to receive more than 168,000 long-range bursts, which allowed us to successfully recover 9,511 complete time stamps from 32 individual bats. To mitigate the impairment by interfering

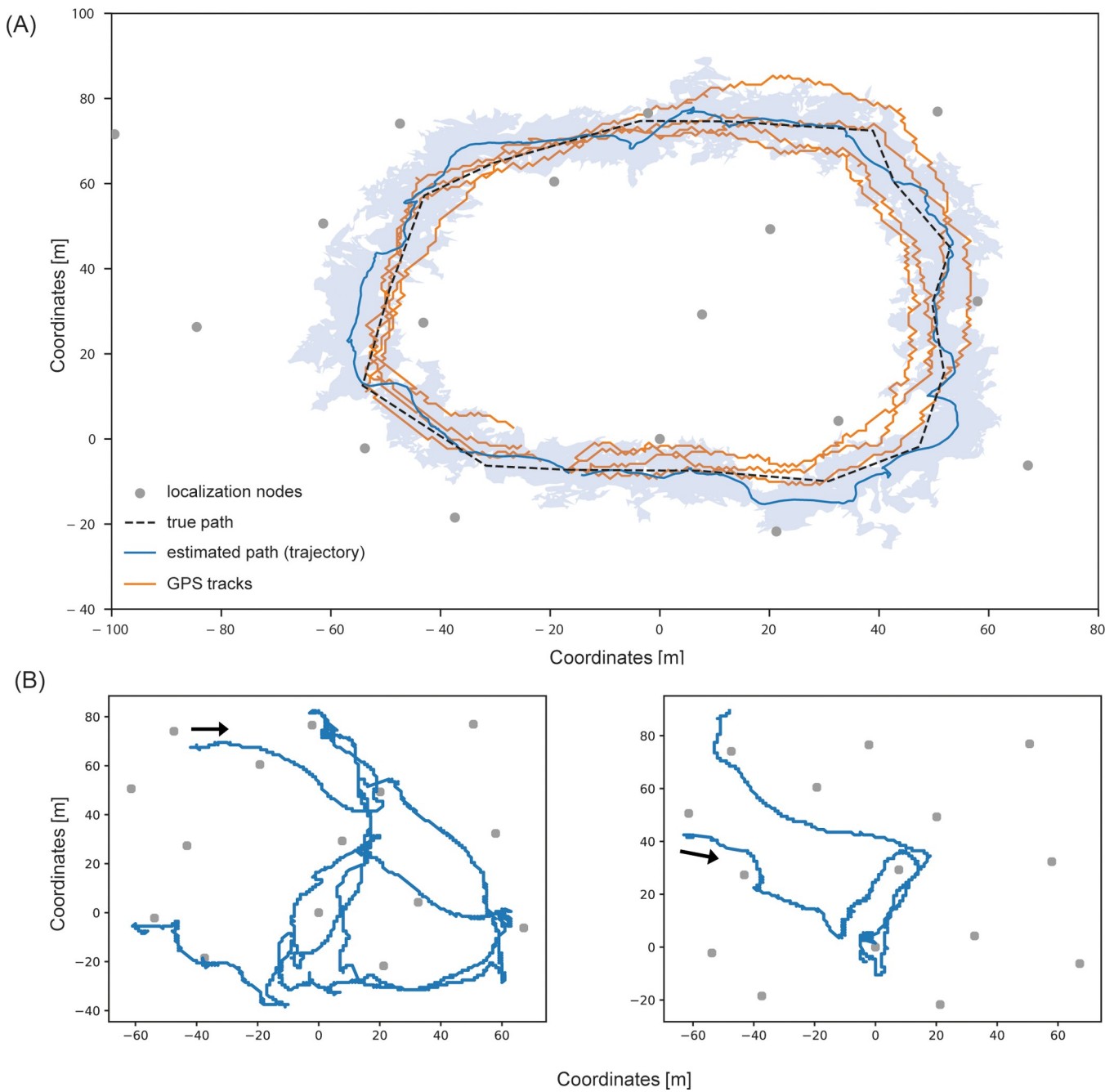

**Fig 3. Tracking bat movements in a forest.** (A) Tracking grid in a deciduous forest of Forchheim, Germany, consisting of 17 localization nodes (gray dots) covering an area of approximately 1.5 ha. Dashed black line: known reference path; blue line and blue shading: estimated path and average localization error obtained by the presented wireless biologging system; yellow lines: 4 individual GPS tracks. (B, C) Estimated flight trajectories of a tagged mouse-eared bat during foraging on August 2nd and 5th. Data and code used to create this figure have been archived by GFBio (https://doi.org/10.7479/vd6t-7a92). GPS, global positioning system.

transmission, 1 complete long-range telemetry packet is split over 24 single-burst transmissions. At the long-range receiver, 24 subsequent bursts are merged to 1 actual long-range packet containing the ID and the bat's time reference. This transmit scheme assures that the mobile node's transmit module is only activated for a short time period, avoiding stress on the

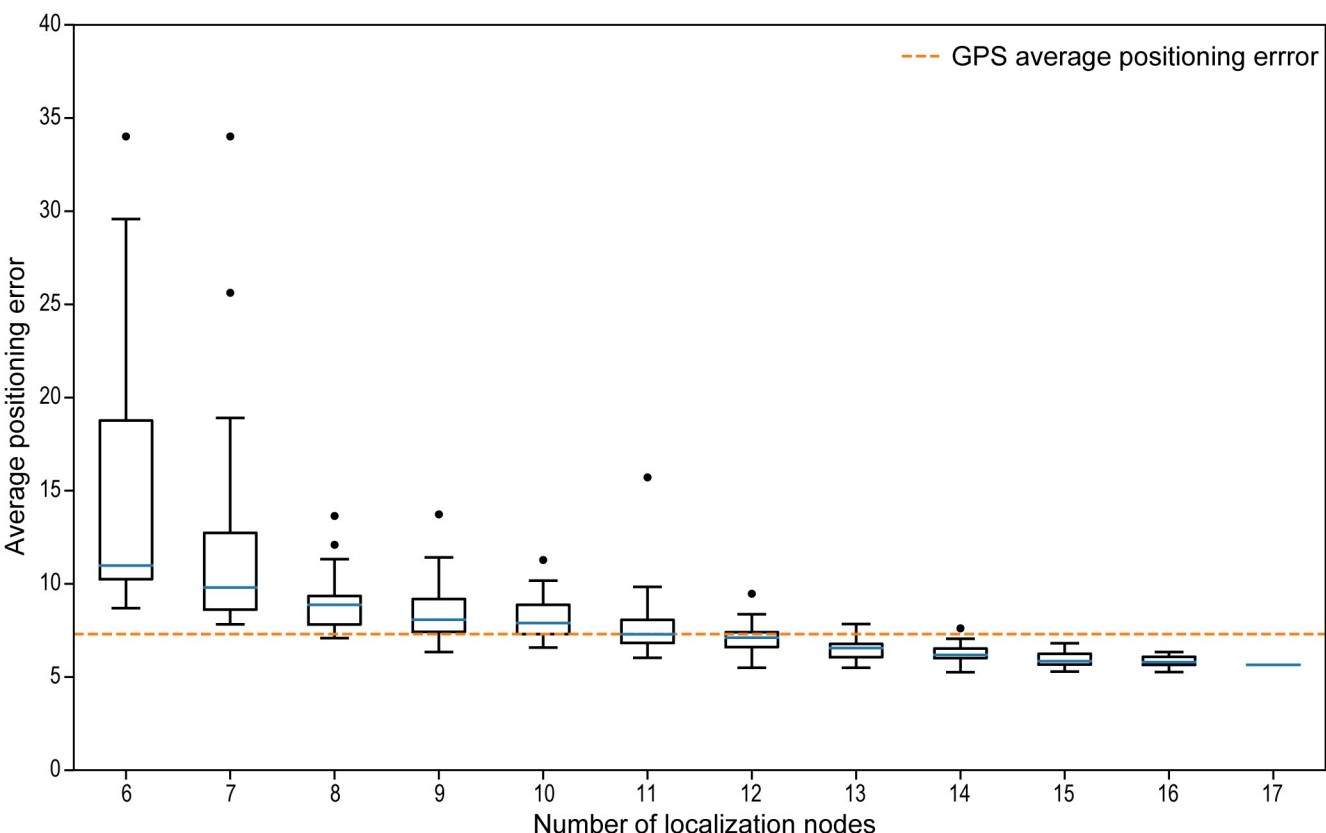

**Fig 4. WBN tracking performance versus GPS tracking.** Localization errors of a reference path of approximately 300 m by the WBN are shown for different numbers of tracking nodes (6–17) in a deciduous forest of approximately 1.5 ha area. The average positioning error of 4 tracks of a heavy-duty commercial wildlife GPS tracker is shown for comparison by a yellow dashed line. Data and code used to create this figure have been archived by GFBio (https://doi.org/10.7479/vd6t-7a92). GPS, global positioning system; WBN, wireless biologging network.

batteries and hardware. In addition, it avoids interference of other channels by a time-frequency hopping pattern in transmission. Instead of a complete package loss, only a fraction of the collection of bursts might be corrupted, which can be reconstructed by means of error-correction codes at the receiver side. It is only because of this specialized telegram-splitting technique [20] that a long-range transmission under extreme power restrictions and vastly occupied frequency channels becomes possible.

Reception of long-range data should perform best when the tagged bats move in open airspace. However, we recovered a considerable number of these long-range bursts while bats were inside their roosts during the day. We mapped 563 long-range bursts received during the day to the known roosts of the bats, allowing us to measure the transmission distance. Fifty-seven long-range bursts from 4 bats inside their roost were recovered over distances of approximately 4.2 km (between 2 roosts and the receiver at the cogeneration plant). We recovered 506 long-range bursts from 5 tagged bats at distances between 667 and 819 m (between the receiver at the retirement home and one of the aforementioned roosts and an additional roost). Burst retrieval over distances of more than 4 km was surprising. Theoretical calculations predicted transmission distances of about 5 km assuming barrier-free transmission [21]. In the field, however, signals had to pass first through the wooden wall of the tree roost and, second, through the forest's vegetation, which should greatly reduce transmission distance.

Data recovery is a major challenge in automated lightweight tracking systems. Signal transmissions inherently suffer from limited transmission power under heavy losses because of distance, shadowing, and other interfering signals. Remote downlinks, e.g., per Global System for Mobile Communications (GSM), add considerable weight in the form of circuitry and battery [1]. Many lightweight trackers must therefore be retrieved, or energy harvesting must be used to counterbalance the expenses for remote data download [7,22–24], again adding weight for the required hardware components. When tagged animals move on predictable scales, energy-saving methods like transfer via very high frequency (VHF) or radio modems may be an option to receive data over distances of hundreds of meters to a few kilometers [25]. Our scenario explores options to decrease the energy expense for downloading stored data to a negligible proportion of the overall energy budget (compare Fig 5). For data download over short distances of approximately 100 m, we accumulate and preprocess data on board and use sophisticated communication protocols that maximize data package reconstruction while minimizing energy demand [26,27]. The above described long-range telemetry mode provides an option for robust transmission of small amounts of data at a low rate without the expenditure of additional energy because of the hybrid modulation of the signal. In comparison, other long-range systems for biologging, such as LoRa [12], enable higher transmission rates. However, the bidirectional communication between transmitter and receiver strongly increases the energy demand on the mobile node.

## Sensor node energy consumption and lifetime

A major strength of WBNs is the ease of adjusting parameters such as sampling rate and in turn energy consumption. These adjustments can maximize runtime for a given battery capacity, or alternatively, maximize sampling rate to obtain higher resolution data. To investigate the impact of the different software task parameters on runtime, we derived a model for energy consumption. We computed examples of runtimes of mobile nodes for 2 battery capacities and different parameter settings (Table 1). For example, the increase in energy consumption when tracking 2 to 4 hours per day can be compensated by extending the mobile node beacon intervals. We achieve runtimes of at least 5 days using a 12 mAh battery (corresponding to a 1 g mobile node) even at the shortest beacon intervals of 2 s (active mode) and with 2 h of high-resolution tracking per day. Depending on the parameter settings, we achieve runtimes of up to 13 d using the smaller battery and 25 d using 22 mAh (Table 1).

Fig 5 illustrates the energy consumption of the different software tasks on the mobile nodes of the 6 different scenarios described in Table 1. When localization is disabled (i.e., only proximity sensing), sending out beacons to wake up other mobile nodes to initiate meetings strongly drives the energy demand (Fig 5A). Therefore, modifying the beacon intervals of the mobile node has the highest impact on runtime (Fig 5B and Table 1). When localization is enabled, tagged animals send localization packages whenever they enter the tracking grid. The high duty cycle of sending localization packages (8/s) strongly decreases the runtime (Fig 5C–5E and Table 1). At active/inactive beacon intervals of 2 per 10 s, a daily localization period of 2 h decreases the overall runtime by 10.8% (Fig 5A and 5C and Table 1). At 4 h of localization, the energy demand for localization dominates the overall energy consumption, in particular at high beacon intervals of mobile nodes.

Model-derived runtimes were compared with empirical runtimes from field tests on noctule bats (*N. noctula*), in which mobile nodes were either powered by a 12 mAh battery or a 22 mAh battery, resulting in masses of mobile nodes of 1.1 to 1.9 g depending on housing. The average runtime was 148 h (max. 209 h) for the small battery, whereas the model predicted 151 h. For the larger battery, average runtime was 280 h (max. 426 h) with a predicted average

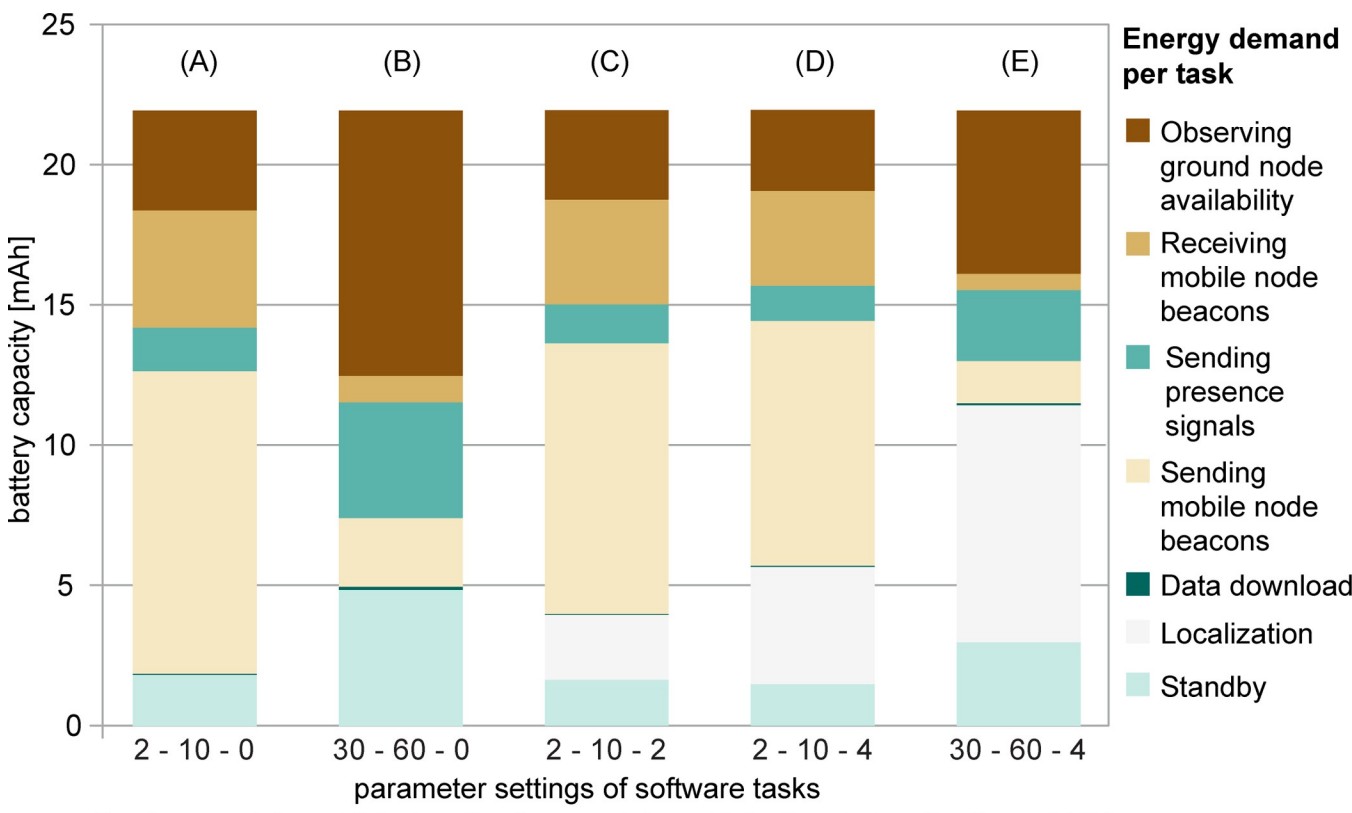

**Fig 5. Energy distribution of software tasks of a mobile node powered by a 22 mAh battery.** Energy demand per software task depends on parameter settings for active/inactive beacon interval (s) and amount of time an animal spends in the localization grid (h). The energy demand is shown for the 7 major software tasks. Zero time in the localization grid (A, B) refers to a pure proximity sensing scenario. Data underlying this figure have been archived by GFBio (https://doi.org/10.7479/vd6t-7a92).

value of 277 h. For both batteries, the predicted values were very close to the observed (1.8% overestimation and 0.7% underestimation, respectively) indicating that the model is a reliable tool for designing a field study.

**Table 1. Estimated runtimes of mobile nodes for 2 battery capacities of 12 or 22 mAh inferred by an energy model for mobile-node runtime.** Although the model comprises 7 energy consuming tasks, the shown runtimes are based only on varying beacon intervals of mobile nodes and localization time (i.e., animal is within the localization grid). For mobile node beacon intervals, 2 operation modes are possible, depending on whether an animal is within reception range of a ground node (inactive mode) or not (active mode).

| Mobile-node beacon interval (s) | | Time inside tracking grid per day (h) | Estimated runtime (h) for a battery capacity of 12 / 22 mAh |
|---|---|---|---|
| if absent from ground node (active mode) | if near ground node (inactive mode) | | |
| 2 | 10 | 0 | 151 / 278 |
| 10 | 30 | 0 | 321 / 589 |
| 2 | 10 | 2 | 135 / 248 |
| 10 | 30 | 2 | 257 / 471 |
| 30 | 60 | 4 | 247 / 454 |

## Discussion

In past years, developments in high-performance proximity sensing using significantly larger animal-borne tags [28] and improvements in ground-based high-resolution tracking in low-clutter desert environments [10] have previously pushed the boundaries of what was technologically feasible. Here, we take the next step by combining these functionalities while keeping the tag mass at 1 to 2 g. Adhering to the 5% rule [29], even animals weighing as little as 20 g can be tagged with this system. These smaller species make up a large proportion of birds and mammals (see Fig 3 in the work by Kays and colleagues [1]), and WBNs will give researchers new capabilities to address a wide range of questions in animal behavior and ecology. Our adaptive and scalable system design provides great flexibility to tailor such a system to meet different species- and study-specific requirements. The design of the mobile node allows one to add multiple functionalities beyond the ones presented here, such as accelerometers, magnetometers, or even an on-board electrocardiogram (ECG) sensor [30].

An early example of automated tracking of small-bodied animals on a limited geographic scale was the automated radio-telemetry system (ARTS), a system that was installed on Barro Colorado Island, Panama [31]. This 6-year endeavor in the 2000 s already highlighted the promising opportunities offered by WBNs—scalability, remote reconfiguration, full automation, and low-cost tags. Yet, a rather low positioning accuracy (approximately 50 m) and restricted coverage were limiting factors [31]. Most of today's solutions for automated tracking of small animals such as songbirds, bats, or rodents perform best at larger geographic scales or in open habitats. For example, current versions of 1 g GPS loggers are suitable to explore seasonal large-scale movements [22,24]. However, they cannot reconstruct flight paths in a complex environment, because they only provide around 100 fixes. Furthermore, the physical devices must be retrieved for data recovery and satellite reception suffers within vegetated areas. Alternatively, reverse GPS can track small animals much more energy-efficiently and at much higher temporal and spatial resolution by measuring time of flight at ground-based receiving stations [10]. However, time-of-flight measurements are inherently affected by vegetation and perform best in open areas. Therefore, we combined signal strength measurements (including angle of arrival (AoA) estimates) from 2 frequency bands and probabilistic multipath mitigation [32] to create a system that is robust to multipath propagation and thus performs well in complex environments. Common but costly measures to resolve multipath propagation are large aperture antenna arrays for AoA tracking or large signal bandwidth for time-of-flight tracking. However, the future of animal tracking will most certainly center on low-cost, ultra-low power integrated circuits, which are currently experiencing a noticeable push because of their broad applications in Industry 4.0 and 5G. This technology has the potential to dramatically boost the capabilities of biologging devices.

Contact networks of small-bodied animals have received increased attention in past years and are most commonly built from passive integrated transponder (PIT)-tagged animals that were observed to be feeding or sleeping at the same site at the same time [33–35]. Later developments for direct encounter logging were able to log associations independently of the locality. However, because of the high energy demand for the permanently active receiver, these sensors were either quite large and heavy [11] or had short runtimes of less than 24 h [36]—major shortcomings for applications in small-bodied species. We show that the use of wake-up receivers and adaptive operation paired with novel wireless communication protocols dramatically reduce the energy demand of such wireless sensor tags. Levin and colleagues [36] achieve approximately 15 h logging time at 20 s interpulse intervals using a 1.3 g mobile node, which is similar to our simulated scenario of 10 s active and 30 s inactive interbeacon intervals, which allows for a sampling period of 321 h using a 1 g mobile node. Extending interbeacon intervals

for the sake of prolonged runtimes always bears the risk of missing short but biologically relevant encounters. For example, at a 20 s interbeacon interval, we would have missed many encounters among flying bats, which were crucial to reveal evidence for maternal guidance in noctule bats [18]. We believe that direct encounter logging, or more precisely, proximity sensing will enable diverse research in the future, because this approach creates large data sets, with additional sensor data providing the behavioral contexts that can close the gap between social patterns and their underlying processes [2].

Ongoing work on ultra-low power sensor networks not only targets animal tracking systems but a variety of general solutions for the "Internet of Things". Energy efficiency is not only a question of hardware circuit design but also of how to interact across all relevant layers in the node's software stack (i.e., application and operating system). On the mobile node, the interaction aspect between communication layers (e.g., application and media access control (MAC) layer) concerns the placement of certain functions (e.g., retransmissions) within the entire software hierarchy [37] such that energy-efficient operation is not affected by unnecessary functional redundancies. Besides, cross-layer designs that optimize the timing of communication processes and make them deterministic at least within limits [38] form the software-engineering basis for an overall energy-aware system approach. In our scenario, energy-efficient and reliable communication between nodes are cross-cutting concerns, because failed communication attempts lead to additional overhead for retransmissions. Very promising examples for low-power communication initiation include novel selective wake-up receivers [39], which allow the small tags to enter sleep modes in the nanoampere range rather than constantly operating in the micro- or milliampere range. Selective wake-up concepts can be used to wake up dedicated recipients of a message (or a selected subset thereof) instead of waking up all systems in communication range. Integrated into the animal tracking nodes, this could enable the next quantum leap on low-power operation.

The alternative to making the receiver operate on a lower energy budget is to make the communication more reliable. Recent advances in integrating coding for forward error correction into such lightweight systems show very promising results, e.g., using erasure codes [26]. Ultrareliable communication protocols currently used in 5G networks can also be applied to localization nodes, which are used for quasilive tracking in the bat tracking scenario. For example, the ground network can be used as a distributed antenna array, which allows the use of smart decoding algorithms for very weak communication signals to further optimize data recovery [40].

## Conclusions

There is no single best method for tracking animal behavior (compare Table 2). Passive RFID technologies or barcodes allow monitoring presence and association of animals at known sites or in lab setups at low cost and in the long-term. Satellite-based localization will remain the method of choice to monitor large-scale movements such as migration or to explore unpredictable events such as nomadism [1,41]. Lightweight GPS tags may even be used to infer social encounters in small vertebrates, but the tags have to be retrieved to access data—a major risk during data collection (Table 2). Simultaneous location tracking of several individuals and estimating interindividual interactions by postprocessing becomes possible by automation of traditional VHF tracking [42]. Single-board computers and software-defined radios now allow for building low-cost open-source solutions that can track a multitude of VHF-tags simultaneously [43]. Similarly, we believe that WBNs like the one presented here will greatly benefit biologging of small animal species that move over smaller and more predictable spatial scales, especially inside habitats where signal transmission is constrained. However, prior knowledge

**Table 2. Overview of tracking systems to track locations and/or encounters in small vertebrates (animal-borne tag $\leq$ 2.5 g).**

| System | Localization quality | Encounter detection | Spatial scale | Data access | Tag mass and costs | Strengths (+) and Limitations (−) |
|---|---|---|---|---|---|---|
| WBN (this study) | • 8 triangulations/s in 2 frequency bands<br>• highly resolved trajectories | • direct proximity sensing among animal-borne nodes (configurable pulse rates; up to 1/s) | • proximity sensing: global<br>• tracking: few hectares (scalable) | • remote short- and long-range download | • 1–2 g<br>• $400–$500/tag<br>• $700/base station<br>• $1,600/high-resolution tracking station | • signal strength gives encounter context (+)<br>• tracking robust to multipath scattering (+)<br>• tags are reconfigurable during operation (+)<br>• system requires thorough calibration, postprocessing of tracking data is laborious (−) |
| Pathtrack nanoFix GEO-Mini[1] [46] | • 640 locations<br>• interfix intervals 30 s to 18 h | • indirect by co-localization during postprocessing | • global | • tag retrieval (often by additional VHF tag) | • 1.7–1.9 g + optional VHF tag<br>• £315–£420/tag | • data collection schedulable (+)<br>• simple handling (+)<br>• fix acquisition very reliable (+)<br>• short operation time at high fix rate (−) |
| Vesper GPS logger platform[2] [47] | • 4 h of GPS logging at 1 fix per10 s and 100% audio recording<br>• fix rates configurable between 1 per s to 1 per h | • direct encounter detection by acoustic recordings of nearby individuals | • global | • tag retrieval (often by additional VHF tag) | • 2.5 g + coating + optional VHF tag<br>• $350–$400/tag<br>• $860 docking station | • short but schedulable runtime (−/+)<br>• encounters are spatially resolved (+)<br>• encounters rely on vocalizations and partner identity remains unknown (−) |
| ATLAS[3] [48] | • usually 1 fix per 4–8s (multiple fixes per s possible) | • post hoc analysis of distance among tracked individuals | • e.g., 10 × 10 km using 9 receiver stations (scalable) | • stored to server; accessible via internet | • 0.9 g (battery and casing included)<br>• <€100/tag<br>• approximately €5,000/receiver station plus server | • live-tracking for experimental triggers (+)<br>• modular and scalable (+)<br>• installation, maintenance and data-processing laborious (−)<br>• power connection and internet required (−)<br>• best performance only in aerial species (−) |
| ARTS-grid[4] [42,49,50] | • 1 position/min (but scalable)<br>• positions base on 50 subsequent samples (3 samples/s) | • post hoc analyses:<br>  ◦ attraction/avoidance (step selection functions)<br>  ◦ dyadic proximity from positions (within 7 m Y/N) | • up to 1 ha using 1 grid (4 antennas, 1 receiver unit)<br>• scalable using several modules | • stored at receiver station | • 0.3 g/tag + collar + casing (0.8 g total)<br>• approximately €150/tag<br>• €6,000–8,000 €/grid | • established technology (VHF) (+)<br>• modular, scalable design (+)<br>• portable setups, easy handling (+)<br>• simultaneous tracking of tags requires multiple receivers (−) |
| Encounter Net[5] [36,51,52] | • Presence near base stations (triangulation theoretically possible) | • direct proximity sensing among animal-borne nodes (configurable pulse rates; 1 per 20 s in the work by Levin and colleagues [36]) | • proximity sensing global<br>• presence detection near base stations | • download to local base stations | • 1.3 g<br>• $250/tag<br>• $350/base station | • signal strength gives encounter context (+)<br>• short but schedulable runtime (approximately 15 h of logging in the work by Levin and colleagues [36]) (−/+)<br>• currently unavailable (−) |

(*Continued*)

**Table 2.** (Continued)

| System | Localization quality | Encounter detection | Spatial scale | Data access | Tag mass and costs | Strengths (+) and Limitations (−) |
|---|---|---|---|---|---|---|
| Passive RFID[6] [34,53] | • identification of individuals at reader station (typically at feeders or roosts) | • indirect encounters from a temporal sequence of detections at a reader | • hectares to square kilometers | • stored at reader station | • approximately 0.1 g • approximately €2/tag • readers starting at €50 | • ideal for wild long-term studies (+) • encounter data are spatially resolved (+) • data collection only at reader station (−) • natural resources difficult to monitor (−) |
| Barcodes[7] [54] | • visual identification at high spatiotemporal resolution; e.g. up to 30fps from video | • direct visual observation of encounters, proximity, and behavioral context | • e.g., 90 × 50 cm with one setup in the work by Alarcón-Nieto and colleagues [54] (but easily scalable) | • stored on recording device | • 0.27 g • €0.1–€0.2/tag | • automated categorization of interactions and experimental triggers possible (+) • restricted applicability in the wild (−) |

Information partly obtained by personal communication:

[1]J. Kohles,

[2]Y. Yovel,

[3]M. Roeleke,

[4]J. Eccard,

[5]I. Levin,

[6,7]D. Farine.

**Abbreviations:** ARTS, automated radio-telemetry system; ATLAS, advanced tracking and localization of animals in real-life systems; GPS, global positioning system; RFID, radio-frequency identification; VHF, very high frequency; WBN, wireless biologging network.

on the study subjects is often useful to ensure individuals regularly visit tracking grids or come close to download stations. High-tech tracking solutions may also increase the workload for maintenance and postprocessing and the equipment may be more vulnerable to external factors, such as increased humidity in tropical ecosystem. Initial purchase costs may also be higher compared to older systems, but this difference might be compensated by the gain in data quantity and quality, the ability to reuse components (e.g., mobile nodes), and the high degree of automation, which can in turn reduce labor costs. The homologies between applications in mobile communication and biologging (e.g., bluetooth low energy for communication among mobile nodes [12]) will boost and cheapen the development of WBNs. Experimental setups including automated triggers (e.g., acoustic playbacks or other sensory cues) can be integrated with direct proximity sensing, creating exciting research opportunities. Such setups will allow studies on the effect of social network dynamics on phenomena such as transmission of social information [34] and pathogens [44] and key ecosystem functions such as pollination and seed dispersal [45].

## Methods

### Ethics statement

**Work on vampire bats.** Our protocols adhered to the following guidelines: (1) The US Government Principles for the Utilization and Care of Vertebrate Animals Used in Testing, Research, and Training, developed by the Interagency Research Animal Committee and adopted in 1985 by the Office of Science and Technology Policy; (2) The Animal Welfare Act, 7 United States Code (USC) §2131 et. seq., and the regulations promulgated thereunder by the US Department of Agriculture (USDA); (3) Public Health Service (PHS) Policy on Humane

Care and the Use of Laboratory Animals, August 2002, for all PHS- or National Science Foundation (NSF)-supported activities involving vertebrate animals. All experiments were approved by the Smithsonian Tropical Research Institute Animal Care and Use Committee (#2015-0915-2018-A9 and #2017-0102-2020) and by the Panamanian Ministry of the Environment (#SE/A-76-16 and #SE/AH-2-17).

**Work on noctule bats.** Our protocols adhered to the following guidelines: (1) The Directive 2010/63/EU of the European Parliament and of the Council of 22 September 2010 on the protection of animals used for scientific purposes; (2) Bundesnaturschutzgesetz (BNatSchG); (3) Tierschutzgesetz (TierSchG); (4) Verordnung zum Schutz von zu Versuchszwecken oder zu anderen wissenschaftlichen Zwecken verwendeten Tieren (Tierschutz-Versuchstierverordnung—TierSchVersV). All necessary permits were obtained from "Senatsverwaltung für Umwelt, Verkehr und Klimaschutz" in Berlin (I E 222/OA-AS/G_1203) and "Landesamt für Gesundheit und Soziales" in Berlin (I C 113-G0008/16).

**Work on mouse-eared bats.** All experiments were approved by the government of Upper Franconia (55.1–8642.01-15/13) and by the government of Lower Franconia (55.2-DMS-2532-2-181).

## The WBN hardware, software, and functionality

Fig 1 is a schematic overview of the presented WBN, which has been developed within the BATS-initiative (for "Betriebsadaptive Tracking-Sensorsysteme," a German short title for "dynamically adaptable positioning of bats using embedded communicating sensor systems"). In order to study its performance, we empirically evaluated the 3 major functions of the system: proximity sensing (Fig 1A), high-resolution tracking at local scales (Fig 1B), and long-range telemetry (Fig 1C). See glossary (S1 Table) for definitions of terms.

## Proximity sensing

Any given mobile node dyad generates meetings whenever it comes within reception range (5–10 m depending on the environment). The animal-borne mobile node consists of a 22 mm × 14 mm Flex PCB circuit board, which is populated with a central System-on-Chip (EFR32, Silicon Labs) containing an ARM Cortex-M4 core and 2 radio frontends for 868/915MHz and 2.4GHz (S1 and S2 Figs). The transmitter in the sub-GHz frontend periodically sends mobile-node beacons, a signal that contains a wake-up sequence. The rate of beacons is configurable (see below). A low-power wake-up receiver on the mobile node triggers the conventional receiver to receive incoming information on the ID whenever a mobile-node beacon is received from another mobile node. Subsequently, a meeting is created between the communicating mobile-node dyad (Fig 1A left). Whereas a conventional receiver draws a relatively high current in receiving mode waiting for incoming packages, a wake-up receiver achieves this functionality with a low current (yet, at cost of sensitivity and performance). When no further mobile-node beacons are received from the meeting partners for 5 interbeacon intervals, the meeting is closed and stored to memory along with the ID of the meeting partner, meeting duration, maximum received signal strength, and a relative timestamp. The mobile node contains both persistent and volatile random-access memory for data storage.

The conventional receiver of the sub-GHz frontend is periodically activated to observe the presence of a ground node (at a fixed interval of every 2 s), which is indicated by a ground-node beacon, periodically broadcast by the transmitter of each ground node (Fig 1A right). The transmitter supports several configurations defining the main purpose of the ground node and enabling location-dependent adaptive operation of the WBN. (i) A download-dedicated ground node broadcasts a signal that enables transmitting mobile-node data based on a

customizable threshold of signal strength received at the mobile node. (ii) A tracking-dedicated ground node positioned within the grid of localization nodes for high-resolution tracking broadcasts a signal that activates the 2.4 GHz frontend in addition to the sub-GHz front end on the mobile node, transmitting "localization packets" at a rate of 8 packets per second. (iii) A presence-detection-dedicated ground node triggers the transmission of "presence signals" by a mobile node and stores incoming signals that can be used to determine presence/absence of tagged individuals (presence at resources or at sleeping sites). Combinations of functionalities (i–iii) may be used in a single ground node if desired (e.g., a tracking-dedicated ground node can also trigger data download). Incoming mobile node data is received by the ground node and stored by a Raspberry Pi (Raspberry PI Foundation, Cambridge, UK) to a SD card along with the ID of the transmitting mobile node and the receiving ground node, respectively, and a timestamp, which is provided by a GPS unit. The Raspberry Pi also hosts a WiFi allowing the user remote data access (software and hardware design can be downloaded at http://coll.mfn-berlin.de/data/10.7479/z5ym-kx58).

Visualization of proximity sensing data is facilitated by the custom-made software "meeting splitter" (see Fig 2). For each specified mobile node ID, the current meeting partners are projected onto a discrete time axis (1 s resolution). We specified a configurable time window around each point on the time axis (5 s in the case of Fig 2). All ongoing meetings, which overlap with the window around the respective point in time, are included in the set of associated bats at this particular point in time. The result per bat is a set of associated bats per each second in the data set. A subsequent automated analysis classifies each meeting as inside or outside the roost, depending on the number of simultaneous meeting partners (not applied in this manuscript).

## Received signal strength-based high-resolution tracking

Localization nodes perform field strength measurements, which are collected by WLAN and are processed by a PC including a file system whenever animal-borne mobile nodes enter the localization grid (Fig 1B; localization nodes collect localization packets from mobile nodes; the transmission is triggered by a ground node). Each localization node comprises a software-defined radio (consisting of a radio frequency frontend, a highly integrated analog-to-digital converter, a field-programmable gate array and a microcontroller) and 2 receiving antenna gain patterns each with 2 main lobes (Fig 1B, red and blue pattern, respectively). The bilobed shape indicates the directional sensitivity of the antenna, and the direction of each lobe represents its maximum in sensitivity. The red pattern is rotated by 90° compared to the blue pattern, and both are simultaneously used to estimate the AoA of the localization packages transmitted by the animal-borne mobile nodes. The difference in received signal strength of the 2 patterns relates to AoA: If the difference—received signal strength of the blue pattern minus received signal strength of the red pattern—is maximum, the wave front impinges on the localization node either from east or west; if the received signal strength difference is minimum, the direction of arrival is north or south. Accordingly, there are 4 options for the AoA if the difference is zero: northeast, northwest, southeast, or southwest. These ambiguities are resolved by fusing measurements of several localization nodes.

This design allows us to exploit not only error-prone absolute field strength measurements [55] but also fail-safe AoA. These are not affected by faulty propagation laws or shadowing effects, because both error sources disappear when forming the received signal strength differences. The angular resolution of the AoA estimates improves with increasing ambiguity of the antenna pattern designs. However, more localization nodes have to be in reach to resolve the

ambiguity [56]. During the Forchheim field trial, we collected up to 272 angle estimates per second when all 17 localization nodes were in reception range.

To further improve localization accuracy, we exploited 3 sources of information ((a) model-based Bayesian positioning, (b) frequency diversity, (c) retrodiction), which increase robustness against multipath propagation. This effect complicates the positioning process, in particular in structurally complex environments because wave fronts impinge a localization node out of the different directions of multiple reflectors (e.g., surrounding vegetation). The information sources to counteract multipath-related adverse effects are described in the following:

**Model-based Bayesian positioning.** Because of the nature of multipath propagation, a stochastic model can be devised to characterize the resulting spread in the AoA estimates [32]. This AoA measurement model can be incorporated into the likelihood function of the recursive Bayesian positioning process, e.g., based on a Kalman filter or a related grid-based estimation filter [57]. The recursive estimation process yields a probability distribution characterizing where the bat may be, considering propagation characteristics from a local channel model [58]. All measurements are fused during the recursive process taking into account a movement model reflecting the flight characteristics of a bat (e.g., max. flight speed). The better the agreement of the various AoA estimates, the more pronounced the positioning probability distribution.

**Frequency diversity.** Multipath propagation leads to frequency-dependent fading. We therefore measured field strength not only on the primary far-reaching carrier frequency at 868 MHz but also on a secondary carrier frequency at 2.4 GHz. On both carrier frequencies, wave forms comprising several subcarriers are employed to enhance the field strength based AoA estimation process. Because of the large carrier frequency separation (>1.4 GHz), frequency-dependent fading effects are decorrelated even if multipath time-of-flight differences are minor, i.e., in the range of a few meters, which corresponds to our accuracy level.

**Retrodiction.** If we do not have to estimate the position of a particular bat in real time, we can exploit all measurements of a bat to estimate a complete trajectory. Forward-backward filtering enhances estimation quality considerably, yielding a positioning quality in the range of 4 m ($1 - \sigma$). Performance limits of field strength based positioning have been discussed in depth [56,59].

We evaluated the trade-off between tracking grid density and localization quality for the Forchheim setup, which comprised 17 localization nodes. In particular, we asked, how many localization nodes are required to obtain localization quality comparable to heavy-duty GPS tracking? We selected subsets of 6 to 16 out of the 17 localization nodes in order to observe the decrease in positioning accuracy with decreasing grid density. Trajectories including standard deviation were estimated for each subset of localization nodes. Seventeen configurations were calculated for the grid consisting of 16 nodes (all possible subsets of the full grid) and 25 unique, randomly chosen subsets for all remaining grid configurations (6 to 15 nodes, respectively) to obtain average errors for the given number of nodes (see Fig 4).

## Long-range telemetry

Our long-range telemetry approach aimed at transmitting long-range bursts from mobile nodes over distances of up to several kilometers—much longer distances than our download-dedicated ground nodes would allow for—within the city of Berlin, under harsh shadowing by obstacles (vegetation, buildings, etc.) or in presence of numerous interferes. We periodically transmitted "long-range bursts," i.e., relative time stamps in the form of seconds since mobile node start-up generated by a simple clock counter. These time stamps are crucial for

postprocessing of meetings because they allow accounting for clock drift on the mobile node (depending on the specific aim of the study other data types such as sensor data could be transmitted instead). We embedded the long-range functionality into the existing modulation scheme using a hybrid phase-alternating modulation on top of the pure amplitude-modulated wake-up sequences of the mobile-node beacon [60]. As a consequence of the extreme energy limitation of the mobile node, we ensured the required signal to noise ratio (SNR) by counterbalancing the rate and the desired transmission distance. The combination of the hybrid modulation, the channel encoding procedure [21,60], and the "Telegram-splitting" technique [20] enables an ultra-low power long-range transmission without additional expenditure of energy. The long-range bursts were received at 2 long-range receivers, which were deployed on exposed sites (rooftops) at distances of approximately 200 to 1,800 m (retirement home; 52˚27'13"N 13˚30'19"E) and 3,300 to 4,500 m (cogeneration plant; 52˚29'18"N 13˚29'36"E) to the proximate respectively ultimate border of the urban forest where the roosts of the tracked bats were located (forest center approximately 52˚27'13"N 13˚29'40"E).

We quantified communication distances in the field, which was possible when a tagged bat occupied a known roost and was simultaneously received by the long-range receiver. We therefore matched time stamps of signals received simultaneously by ground nodes at roosting sites and at long-range receivers. In case of a match, we quantified the distances between roosts and long-range receivers in the R package geosphere using the Haversine function [61]. The empirically assessed communication distances have then been compared to a theoretical model of long-range transmission distances [62]. This model evaluates achievable rate and distance of transmission based on the energy relation of the SNR, presuming the transmission power given by the mobile node's hardware configuration and a desired target payload rate. For simulating the channel characteristics faced by the mobile node, the model comprises parameters like the path loss in dependence of the signal-center frequency, the transmission distance, and receiver and transmitter heights. Environmental influences like attenuation by obstacles, multipath propagation, or unpredictable rotation of the mobile node's rod antenna are incorporated by means of a random variable, stating the superimposed attenuation effects. Based on these assumptions, we were capable of overcoming path losses of over 150 dB for a distance of 5 km and more, under reasonable rates of packet loss [21], thus accomplishing an ultra-robust implementation supporting payload data rates of a few bits per second.

## Sensor node energy consumption and runtime

A crucial aspect for biologging is knowledge on the runtime of the sensor nodes. Static program-code analysis methods of the mobile node are able to determine upper bounds on the nodes' runtime [63]. However, in the context of the presented WBN, precise estimates for the average uptimes of the system proved to be more beneficial for the empirical studies than upper bounds for the lifetime. Consequently, we focused on an energy model to determine the average runtime of the mobile nodes, which is strongly dependent on the tasks executed by the software. Our models are based on measurements of each executed task in combination with empirically determined activity parameters of each task. That way, we ensure highest accuracy for our model. In our setting, 7 different tasks are implemented: (i) standby, (ii) sending mobile-node beacons, (iii) receiving mobile-node beacons, (iv) observing ground-node availability, (v) transmitting data to a ground node, (vi) sending localization packets, and (vii) sending presence signals (see S1 Table for definition of terms).

We determined the runtime by using the specific energy demand for a task and by translating it to an average current draw. With the average current draw and a given battery capacity,

the runtime can be computed as follows:

$$T_{runtimeavg} = \frac{E_{battery} \cdot \eta_{DCDC}}{\sum I_{tasks}}.$$

DCDC represents the efficiency of the DCDC-converter, which is permanently active and consumes energy. Determining DCDC is impractical, because it highly depends on the actual current drain of the application for the entire runtime. For this reason, we assume a fixed efficiency of 0.95, which translates to only 95% of the battery capacity being available for software tasks. This way, losses caused by the DCDC and parasitic discharges of the battery are modeled in a coarse-grained manner.

The idle current during standby is given in a current draw, which does not require any further calculations. The other tasks (e.g., observing ground-node availability, sending localization packets) are executed in predefined cycle times (duty cycle). Based on the measured energy demand and the duty cycle, we calculated an average current draw for each task. The energy demand for each task was measured in the lab with an Agilent DC power analyzer precise source meter. In the case of a localization packet, which is sent every 128 ms, the average current draw can be expressed as follows:

$$I_{localization} = \frac{E_{localizationpacket}}{T_{dutycycle} \cdot V_{supply}} \cdot \frac{T_{localization}}{T_{day}}.$$

The average time spent inside the localization grid $(T_{localization})$ per day is strongly dependent on the species-specific animal behavior and the experimental design. For the calculations presented here, we set the daily localization period to 2 or 4 h, respectively.

Observing a ground node in receiving range is carried out at a fixed duty cycle of 2 s. Here, the task is independent of the behavior of the tracked animal, and the energy demand is calculated as follows:

$$I_{basestationRX} = \frac{E_{basestationRX}}{T_{dutycycle} \cdot V_{supply}}$$

The transmission rate of mobile-node beacons and presence signals is adaptive based on the contact to a ground node (i.e., a mobile node near a ground node at a roost will decrease the duty cycle in comparison to a mobile node on a foraging animal which is not in reception range of a ground node). In turn, the energy demand for sending beacons and presence signals highly depends on the behavior of the tracked study species and the individual animal (e.g., time spent near ground nodes at roosting sites). We therefore used empirical data obtained in the Berlin field test to inform our energy model with realistic averaged parameter values for duty cycles of each task and the amount of transmitted data. We quantified the average time tagged bats spent in reception range of a ground node (inactive mode, decreased duty cycle) versus the time bats spent outside the reception range of any ground node (active mode, increased duty cycle). The common noctule bats in the Berlin field test spent on average 47% of the observation time in the inactive mode, and the energy demand for transmitting beacons and presence signals calculates as follows:

$$I_{beaconTX} = \frac{E_{beaconTX}}{V_{supply}} \cdot \left( \frac{r_{\frac{inactive}{active}}}{T_{inactive}} + \frac{1 - r_{\frac{inactive}{active}}}{T_{active}} \right);$$

$$I_{presenceTX} = \frac{E_{presenceTX}}{V_{supply}} \cdot \frac{r_{\frac{inactive}{active}}}{T_{inactive}}.$$

Receiving a mobile-node beacon depends on the duty cycle at which beacons are transmitted and on the number of mobile nodes in receiving range. During the Berlin field test, we calculated the average number of 2.05 maximum parallel meetings:

$$I_{beaconRX} = \frac{E_{beaconRX} \cdot N_{avgEncounter}}{V_{supply}} \cdot \left( \frac{r_{\frac{inactive}{active}}}{T_{inactive}} + \frac{1 - r_{\frac{inactive}{active}}}{T_{active}} \right).$$

For data download to a ground node, we assumed static energy consumption (the energy demand for sending a data packet highly depends on the size of the packet to be transmitted [27]). The number of data packets to be transmitted is again dependent on the behavior of the tracked animal (depending on how many meetings an individual accumulates). Mobile nodes on common noctule bats sent on average 23.7 packages per hour to a ground node. Thus the current draw can be denoted as

$$I_{download} = \frac{E_{packet} \cdot N_{packetsperhour}}{3600s \cdot V_{supply}}.$$

Based on these calculations, we matched the estimated average runtimes to the observed runtimes during the Berlin field test (based on the last beacon or packet received from each individual tagged bat).

## Adaptive operation, scalability, and reconfiguration

The adaptive operation contributes to energy efficiency. We define location-specific communication schemes on the mobile nodes which are initiated by ground nodes. At a noctule bat day roost, e.g., ground nodes activated the inactive beacon interval where mobile-node beacons for meeting generation were only sent every 10 s. When tagged bats leave their roost and move beyond the reception range of the ground node, the mobile node switches to the active interval, sending a mobile-node beacon every 2 s, which increases the probability to detect also very short meetings in comparison to the inactive rate. Similarly, localization packets, which strongly increase the energy demand, should only be sent when the tagged animal moves within the tracking grid and are therefore triggered by a ground node within the grid.

We can track multiple individuals simultaneously. Our current design allows for the observation of up to a theoretical maximum of 60 individuals. Field deployments containing 11 to 50 tagged bats empirically validated this targeted scalability. The scale of the localization grid can be adapted to the ranges required for experimental setups. Although we tracked mouse-eared bats on approximately 1.5 ha, smaller areas might be sufficient to track, e.g., rodents (bank voles, which showed a density of 40–162 individuals per hectare, have been tracked on less than 0.5 ha using automated VHF telemetry [64]; see ATLAS-system in Table 2). Tracking grids larger than 1.5 ha are certainly possible from a technological point of view. Yet, one has to keep in mind that effort for maintenance (e.g., replacing power sources for localization nodes) scales with the size of the tracking grid.

Because settings for communication schemes are sometimes difficult to pick a priori, we built an option for reconfiguration of so-called soft settings (active/inactive rate of mobile-node beacons, received signal strength thresholds for data download, localization interval, or timeout duration after which a running meeting is terminated, etc.). Four values for every soft setting (e.g., active rate 2 s, 4 s, 10 s, 30 s) can be defined a priori, and during operation ground nodes can be used to trigger a switch between these predefined values at the mobile node.

## Field deployments

The presented system is modular, and hardware setup and software configurations can be tailored to a specific use case. We evaluated the system's performance during 3 major field studies by applying mobile nodes to vampire bats, noctule bats, and mouse-eared bats with body mass of 27 to 48 g, 18 to 35 g, and 22 to 28 g, respectively. Although at least 2 of the 3 major functionalities (proximity sensing, high-resolution tracking, long-range telemetry) have been used in all 3 field studies, we focus on one specific functionality per deployment. The proximity sensing performance was best demonstrated in the vampire bats study because the highest number of individuals was tagged here and group cohesion was high—ideal circumstances to test system performance in the presence of a large number of tagged individuals. We demonstrated high-resolution tracking with mouse-eared bats because individuals of this species regularly revisit their foraging grounds and can therefore reliably be tracked within an area of 1 to 2 ha across multiple days. We assessed transmission distances of long-range telemetry in noctule bats because they frequently switched roosts, and long-range telemetry provided a valuable opportunity to obtain data from individuals that moved to unknown locations. In general, we chose to use bats to demonstrate the high performance of this system because small body mass, nocturnal activity (no sunlight exposure), and flight confront tracking systems with major challenges. A system that is capable of automated tracking of bats may be likely to work for tracking other small terrestrial taxa as well.

## Proximity sensing in vampire bats

We tagged 50 common vampire bats (*D. rotundus;* 44 adult females, 6 subadults) from a colony roosting in a cave tree near Tolé, Panama, to document social networks with high resolution. Field work was conducted during September and October 2017. Mobile nodes were powered by a 22 mAh LiPo battery and housed in a 3D-printed plastic case, resulting in a total mass of 1.8 g. One download-dedicated ground node was positioned inside the roost, and 5 ground nodes were placed on surrounding cattle pastures to detect the presence of foraging or commuting bats.

## Long-range telemetry in noctule bats

We captured and tagged 34 common noctule bats (*N. noctula*; 19 adult females and 15 juveniles) from 2 bat boxes in a nursing colony in an urban forest in the city of Berlin, Germany ("Königsheide Forst") [18]. Mobile nodes were powered by either a 12 mAh or a 22 mAh battery and were housed either in a 3D-printed plastic case or in a fingertip of a nitrile lab glove that was sealed with glue. Total mass varied between 1.1 and 1.9 g depending on housing and battery. We positioned 5 ground nodes underneath known roosts to document the presence of individual bats and to remotely download data. In addition, we set up 2 long-range receivers to evaluate model-based predicted data retrieval over distances of up to 4 to 5 km [21]. This opportunity is particularly valuable to retrieve data of tagged individuals that moved to an unknown roost.

## High-resolution tracking of mouse-eared bats

We captured 11 mouse-eared bats (*M. myotis*) using mist nets set up at ground level in a mature deciduous forest near Forchheim, Germany. When hunting for ground beetles, mouse-eared bats are faithful to their foraging sites for consecutive days. We therefore mist-netted bats at an attractive foraging site (rather than catching them from a roost) in order to track repeated bouts by returning individuals over the course of several days. Mobile nodes

were powered by 22 mAh batteries and housed in fingertips of nitrile lab gloves (total mass 1.4 g). At the capture site, we installed a tracking grid consisting of 17 localization nodes covering roughly an area of 1.5 ha (see Fig 3). Distance between tracking stations varied between approximately 25 and 40 m. The irregular configuration was because of the presence of thick trees. We aimed at positioning tracking stations at least 3 to 5 m away from trees to reduce shielding of the signal. We set up a polygon-shaped reference path for estimating localization errors and determined the true position of the corners using a Leica Robotic Total Station TS16 (positioning error <5 cm). Corners were connected using strings, and we walked either a sensor node or a GPS tracker (Ornitela OrniTrack-15; a 15 g solar powered GPS-GSM/GPRS tracker; maximum logging rate 1 fix per second at a lifetime of approximately 4 h without solar harvesting) along the calibration path and calculated the average localization error based on the obtained tracks.

## Supporting information

**S1 Fig. Animal-borne mobile node for proximity sensing.** (A) Common vampire bat (*Desmodus rotundus*) carrying a mobile node housed in a plastic case; (B) bare mobile node on a quarter US dollar coin for comparison of size. Credits: Sherri and Brock Fenton (A), Peter Wägemann (B).
(TIF)

**S2 Fig. Block wiring diagram of the mobile node.** The assembled node is connected to a lithium polymer battery, which is folded to lie in parallel to the mobile node. A 3D printed plastic case including lid can be used as housing (see S1 Data; a hole for the whip antenna must be added manually). Specific components: DC/DC (DC-to-DC converter) = TPS82740 (by Texas Instruments); NVRAM (nonvolatile random-access memory) = FM25V20A (by Cypress Semiconductor Corp.); Cortex M4 / sub-GHz / 2.4 GHz (microcontroller) = EFR32FG1P133F256GM48 (by Silicon Labs); WuRx (wake-up receiver) = AS3933 (by ams), Switch (single-pole double-throw switch) = SKY13350 (by Skywork Solutions Inc.); antenna 2.4 GHz = AMCA31-2R450G-S1F-T (by Abracon); antenna sub-GHz = whip antenna.
(TIF)

**S1 Table. Glossary.** Description of hardware components, communication, data types, and software tasks considered for the energy model.
(DOCX)

**S1 Data. Case for the mobile node.** STL-file for 3D-printing a plastic housing for the mobile nodes.
(STL)

## Acknowledgments

We thank M. Mutschlechner, M. Nabeel, J. Blobel, and M. Hierold for their contributions within the scope of the research unit. We are grateful to J. Kohles, Y. Yovel, M. Roeleke, J. Eccard, I. Levin, and D. Farine for sharing their knowledge on different biologging technologies. We highly appreciate logistical support during the Forchheim field study by Naturstrom, MTS Meixner Transporte + Service, C. Kreul GmbH & Co. KG, Spedition Pohl GmbH & Co. KG, and Siedlergemeinschaft Lichteneiche. We are grateful to J. Berrío-Martínez, D. Josic, M. Nowak, G. Cohen, L. Günther, H. Wieser, C. Rüstau, B. Peiffer, M. Hammer, F. Oehme, and his BUND group of volunteers for their support during field work. We express our particular thanks to J. and C. Mohr, who have substantially contributed to the success of our field work.

We thank E. Siebert, Museum für Naturkunde Berlin, for making the line work of *D. rotundus* and *N. noctula*.

## Author Contributions

**Conceptualization:** Simon P. Ripperger.

**Data curation:** Simon P. Ripperger.

**Formal analysis:** Simon P. Ripperger, Markus Hartmann, Thorsten Nowak, Sebastian Herbst, Björn Cassens.

**Funding acquisition:** Simon P. Ripperger, Gerald G. Carter, Rachel A. Page, Alexander Koelpin, Robert Weigel, Jörn Thielecke, Jörg Robert, Klaus Meyer-Wegener, Wolfgang Schröder-Preikschat, Rüdiger Kapitza, Falko Dressler, Frieder Mayer.

**Investigation:** Simon P. Ripperger, Gerald G. Carter, Niklas Duda, Markus Hartmann, Thorsten Nowak, Michael Schadhauser, Björn Cassens.

**Methodology:** Simon P. Ripperger, Niklas Duda, Markus Hartmann, Thorsten Nowak, Jörn Thielecke, Sebastian Herbst, Peter Wägemann, Björn Cassens.

**Project administration:** Robert Weigel.

**Software:** Sebastian Herbst, Peter Wägemann, Björn Cassens.

**Supervision:** Rachel A. Page, Alexander Koelpin, Robert Weigel, Jörn Thielecke, Jörg Robert, Klaus Meyer-Wegener, Wolfgang Schröder-Preikschat, Rüdiger Kapitza, Falko Dressler, Frieder Mayer.

**Visualization:** Simon P. Ripperger, Niklas Duda, Thorsten Nowak, Björn Cassens.

**Writing – original draft:** Simon P. Ripperger, Niklas Duda, Thorsten Nowak, Jörn Thielecke, Michael Schadhauser, Peter Wägemann, Björn Cassens.

**Writing – review & editing:** Simon P. Ripperger, Gerald G. Carter, Rachel A. Page, Niklas Duda, Alexander Koelpin, Robert Weigel, Markus Hartmann, Thorsten Nowak, Jörn Thielecke, Michael Schadhauser, Jörg Robert, Sebastian Herbst, Klaus Meyer-Wegener, Peter Wägemann, Wolfgang Schröder-Preikschat, Björn Cassens, Rüdiger Kapitza, Falko Dressler, Frieder Mayer.

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
