## [Editor Report · Decision Letter 0]

28 Oct 2019

Dear Dr Ripperger, 

Thank you for submitting your manuscript entitled "Thinking small: next-generation sensor networks close the size gap in vertebrate biologging" for consideration as a Methods and Resources paper by PLOS Biology.

Your manuscript has now been evaluated by the PLOS Biology editorial staff, as well as by an academic editor with relevant expertise, and I'm writing to let you know that we would like to send your submission out for external peer review.

Please re-submit your manuscript within two working days, i.e. by Oct 30 2019 11:59PM.

Kind regards,

Roli Roberts

Senior Editor

PLOS Biology

---

## [Decision Letter · Decision Letter 1]

27 Nov 2019

Dear Dr Ripperger,

Thank you very much for submitting your manuscript "Thinking small: next-generation sensor networks close the size gap in vertebrate biologging" for consideration as a Methods and Resources at PLOS Biology. Your manuscript has been evaluated by the PLOS Biology editors, an Academic Editor with relevant expertise, and by three several independent reviewers.

You'll see that all of the reviewers are broadly positive, and in light of the reviews, we're pleased to offer you the opportunity to address the comments from the reviewers in a revised version that we anticipate should not take you very long. We will then assess your revised manuscript and your response to the reviewers' comments and we may consult the reviewers again.

IMPORTANT: The Academic Editor notes that the reviewers request explicit qualitative (revs #1 and #2) and quantitative (rev #3) comparisons with other existing technologies (e.g. zebraNet) to better show the novelty, advantages, and disadvantages of your described methodology (Rev #1's suggestion to tabulate the comparison seems helpful). The Academic Editor also agrees with Rev#3's arguments that the BATS acronym is unhelpful in summarising either the content or applicability of the method, and recommends that you avoid it.

Your revisions should address the specific points made by each reviewer. Please submit a file detailing your responses to the editorial requests and a point-by-point response to all of the reviewers' comments that indicates the changes you have made to the manuscript. In addition to a clean copy of the manuscript, please upload a 'track-changes' version of your manuscript that specifies the edits made. This should be uploaded as a "Related" file type. You should also cite any additional relevant literature that has been published since the original submission and mention any additional citations in your response. 

Before you revise your manuscript, please review the following PLOS policy and formatting requirements checklist PDF: http://journals.plos.org/plosbiology/s/file?id=9411/plos-biology-formatting-checklist.pdf. It is helpful if you format your revision according to our requirements - should your paper subsequently be accepted, this will save time at the acceptance stage.

Please note that as a condition of publication PLOS' data policy (http://journals.plos.org/plosbiology/s/data-availability) requires that you make available all data used to draw the conclusions arrived at in your manuscript. If you have not already done so, you must include any data used in your manuscript either in appropriate repositories, within the body of the manuscript, or as supporting information (N.B. this includes any numerical values that were used to generate graphs, histograms etc.). For an example see here: http://www.plosbiology.org/article/info%3Adoi%2F10.1371%2Fjournal.pbio.1001908#s5.

For manuscripts submitted on or after 1st July 2019, we require the original, uncropped and minimally adjusted images supporting all blot and gel results reported in an article's figures or Supporting Information files. We will require these files before a manuscript can be accepted so please prepare them now, if you have not already uploaded them. Please carefully read our guidelines for how to prepare and upload this data: https://journals.plos.org/plosbiology/s/figures#loc-blot-and-gel-reporting-requirements.

Upon resubmission, the editors assess your revision and assuming the editors and Academic Editor feel that the revised manuscript remains appropriate for the journal, we may send the manuscript for re-review. We aim to consult the same Academic Editor and reviewers for revised manuscripts but may consult others if needed.

We expect to receive your revised manuscript within one month. Please email us (plosbiology@plos.org) to discuss this if you have any questions or concerns, or would like to request an extension. At this stage, your manuscript remains formally under active consideration at our journal; please notify us by email if you do not wish to submit a revision and instead wish to pursue publication elsewhere, so that we may end consideration of the manuscript at PLOS Biology.

When you are ready to submit a revised version of your manuscript, please go to https://www.editorialmanager.com/pbiology/ and log in as an Author. Click the link labelled 'Submissions Needing Revision' where you will find your submission record. 

Sincerely,

Roli Roberts

Senior Editor

PLOS Biology

REVIEWERS' COMMENTS:

Reviewer #1:

I enjoyed reading this paper, which was very well written and structured. The Introduction provided a concise but thorough overview of the current state of biologging technology, and clearly identified where the gap/unresolved issues are. The methods that are presented are novel and have the potential to offer a step-change in how movements of small animals can be studied. 

The authors do an excellent job of explaining why this novel system is so beneficial. I do think it would be useful also to have a greater discussion about what some of the potential drawbacks would be. In the Introduction the authors talk about how ICARUS and other existing techniques are limited, but then this is one pilot study, essentially, on one animal group (bats) – will this really work for other systems, and if not, why not, and what systems is it unlikely to be useful for. 

To this end, a summary table that perhaps proposes some different scenarios and animal groups coupled with how this technique could be applied, and whether it would work, would be beneficial to readers. 

So I do think the paper needs some discussion of potential negatives. For example, setting up such a system in the field will not be straightforward, and you will need some prior knowledge of the animals (of interest) movements to set things up. What are the costs involved? How user friendly is it? Many scientists who use biologging technology are conservationists, ecologists etc, and their background will be varied – how simple is this system to implement in reality? 

I think the authors need to enlarge slightly their discussion about the work that originally took place in Panama, using – to some extent – a similar approach. What really didn’t work there, and what has been the massive step change with this method that permits such a system to now work. 

As an end user of biologging technology, rather than someone involved in its design, the methods did make sense to me and the approach logical and correct, but a technical engineer might be better placed to double check the coding details behind the logging technology. 

Minor – weight should be mass throughout; line 59, energy may be better phrased as power? Here also, vertebrates – could be invertebrates too, given the size of some of them

Reviewer #2:

The manuscript provides a really interesting system for advanced, fine resolution tracking of multiple organisms. The contact data generated from the study system was very impressive. The concept of contact information collected and distributed described here is similar to several other approaches that have been described (e.g. zebraNet). It was not clear how novel and different the current system is relative to these other approaches. I would also like to see a better discussion of the costs of building and maintaining the infrastructure for the BATS system – a number of approaches have been proposed and used using local area data transmission through radio frequencies received through towers. How does this differ? How does it compare to other solutions in terms of cost? Without this information it is difficult to gauge the novelty and potential for the BATS system. The reason the ICARUS and GPS systems are alluring is the lack of infrastructure needed to deploy collars in such systems.

1. The number of acronyms used in the main text and figure captions should be reduced to avoid reader confusion. Several were mentioned in the introduction, and then irregularly re-defined throughout the paper. For example MN is redefined in Figure 5 caption (Line 296), but not in Table 1 (Line 278). I recommend removing WBN, RSS, MN, GN and RSSI, because of their sporadic use throughout the paper. 

2. You acknowledge that there is no single best method for animal tracking, and review the limitations and strengths of several tracking technologies in the context of different ecological phenomenon. It would also be useful to see a cost comparison of the high-resolution tracking solutions. At the very least, and mention of the costs of each of the tracking systems used in this paper would be helpful to understand where the BATS system sits relative to other lower-resolution tracking options. 

120: It was unclear what the long-range telemetry in Figure 1 was used for in the tracking system. On line 107 the synchronization of clocks is mentioned. Is that the core use of the long-range telemetry, or is their other contact data being sent when animals are outside of the localization grid? 

200: Can a start point be added for the tracks in (b) and (c)?

203: Should read (b) and (c). 

398: Change “will allow to study the” to “will allow study on the”

Reviewer #3:

[identifies himself as Josh Firth]

This manuscript provides a detailed and useful description of a wireless biologging network system and demonstrates how deploying this technology produces some amazing data detailing bat behaviour. It is very timely given the recent advances of (and demand for) animal tracking systems to provide such fine-scale data. For instance, I’m certain that the data here could be useful for addressing various questions regarding ecology within these bat social systems. The manuscript is very well written and formatted. My comments below are given in the following sections (1) Comments on potential considerations and changes regarding (1a) insights into biology (1b) comparison to other technologies and (1c) cost, and (2) Minor suggestions that may (or may not) be helpful to the authors. 

(1) Considerations and potential changes

(1a) Insight into biology

(1ai) Although the manuscript does an excellent job of describing the technology and its direct uses, it would be very helpful if more explanation of the potential insights into biology that these new improvements could provide (especially insights that might not have been possible to obtain prior to this). For instance, much effort is much into describing, and quantifying improvements such as energy consumption, lifetime, resolution, distance verification, etc etc, but it would be great to know how each of these different improvements will be directly useful for answering different ecological questions. Currently, readers (especially those with little knowledge of uses of tracking technologies) may be left unsure about how scientifically useful these advances are, and which of the advance will be useful for what questions. It would be a shame to miss out on the opportunity here to really show how general and useful advancements in these technologies can be

(1aii) Secondly, also in relation to this general point of providing insights into biology, it currently isn’t entirely clear why the different bat datasets are all used in different ways to look at different components of behaviour. Obviously one of the major ‘selling’ points of this technology is that is should provide great generality in what behaviours can be examined, but the current framework within the MS makes it appear that different datasets have to be used to look at different behaviours. Ideally, the MS would make the most of this amazing data across systems and show that each of the components of behaviour (social interactions, movement angles, distances) can be looked at within each of the systems (rather than separately as currently done).

However, perhaps it is the case that each of these aspects can be looked at in each system, but that this would be far too much extra work here. This is fair enough I think, but it would be good for the MS to clearly state this, and state why each aspect of behaviour was looked at within each dataset specifically here. This would also be a good opportunity to address ‘Comment 1ai’, as then it could be made clear why biologists might want to know more about each of these behaviours (interactions, angels, distances), and why these are important to larger biological questions and why these questions were not easy to answer before. Finally, the bat data here looks excellent, I’m sure a few new insights are provided just by quantifying these systems in this way, it’d be very exciting to here them (or a hint of them) here.

(1b) Other technologies

(1bi) A bit more direct qualitative comparison would be interesting to hear too. Currently, it is fairly heavily focused on comparing to GPS tracking. This comparison is a little bit one-dimensional as GPS are mainly only used by those desiring long distance tracking (and those less interested in other questions) – the specific device considered in the MS wouldn’t be used for for short distances. Nothing too much more is required, but it’d be interesting to here which currently-used technology is most like this one, how it compares to recent developments in lab-based system technologies (e.g. automated high-res tracking of barcodes in zebrafinch systems), how useful this technology might be for lab-based systems.

(1bii) Similarly, some more direct quantitative comparisons to other technologies would be really interesting to see (see point above). Ideally, if quantitative comparisons using simulations could be straight-forward and easy-to-implement then that would be very helpful indeed. For example, it would be nice to know how the results from this high-res tracking technology compares to another similar (but I’m assuming lower-res) technology such as encounternet. Given the sampling rate here, it would be great (and fairly easy?) to resample the data from this technology at a rate equivalent to something like encounternet, and then see how much this changes the inferences about social interactions for instance? 

(1c) Cost

(1ci) The price (cost) of new technologies is a major barrier to many tracking applications, and this study acknowledges the general importance of cost in the discussion L342-347. But, I think it also needs to specifically state the cost of these devices clearly. I think this will be the main question on interested readers’ lips as they hear about this system (it was certainly a burning question for me when I was reading it!). Obviously animal behaviour research funds are often difficult to obtain in most countries, so cost will be the main factor for driving international interest in this. 

(1cii) Along with raw cost of this set-up, some direct demonstration/discussion of cost-effectiveness would be great too. Obviously the system is technically superior to many current options in terms of performance, but how does it compare when cost effectiveness is considered? For instance, there are genuinely very-low cost animal tracking options out there now, with possibilities of building RFID recording devices and tags for 100individuals or so for total cost of around £100. I obviously don’t expect this new system to beat this kind of raw cost, but could it be potentially considered as more ‘cost effective’ than this given the exceptional data in produces? 

(2) Minor suggestions

The following comments might not need to be addressed (or even responded to individually) but I just thought I’d point out a few thoughts I had while reading the MS in case it is helpful.

(2a) I really like figure 1 – I think it looks great. But, what is the meaning of the placement of the trees? I initially thought they were referring to physical barriers and I was trying to determine what the presence/absence of trees in different places meant. But, I now think it’s just to show environment in general and the actual placement has no meaning? If the latter is correct, I suggest having them as background rather than objects.

(2b) I work on using technologies for experiments, not just observations, so I was wondering whether this WBN set-up could be used to manipulate individuals automatically at all? If it can’t be used to carry out automated treatments itself, could it potentially be combined with other low-cost technologies that can e.g. RFID devices used to change predation preception (e.g. Voelkl et al 2016 Sci Reports) or control access to resources (Firth & Sheldon 2015 PRSB)? It’d be exciting to hear some exploratory ‘first-thoughts’ on this topic.

(2c) I get why the acronym BATS was chosen, but I’m not sure it is the best for describing this system. Indeed, lots of different tracking systems are ‘broadly applicable’, so it might be better to have something that describes this tracking system? Also, as this study uses bats (actual bats) as the study system for this tracking technology, this acronym might not end up being the best for quickly representing the ‘broadly applicable’ aspect of this tech (runs the risk of people thinking its focused on bats!).

L310 states ‘experimental design?’ Should this be ‘observational design’?

L313-315 the discussion goes straight into statements about the technology, but ideally it would start with something biologically relevant (i.e.point 1a)

I hope these comments are useful to the authors for this excellently coherent and interesting description of this exciting new technology

Josh Firth (please note, I sign all my reviews).

---

## [Editor Report · Decision Letter 2]

17 Jan 2020

Dear Dr Ripperger,

Thank you for submitting your revised Methods and Resources paper entitled "Thinking small: next-generation Sensor Networks close the Size Gap in Vertebrate Biologging" for publication in PLOS Biology. The Academic Editor and I have now assessed your revisions. 

Based on this assessment, we will probably accept this manuscript for publication, assuming that you will modify the manuscript to address the following points:

IMPORTANT: We and the Academic Editor are concerned that your manuscript currently gives insufficient detail for readers to replicate your Method. This is particularly crucial for a Methods and Resources paper, and in order to comply with PLOS' policy on reproducibility. We therefore request that you supply, preferably as Supplementary files, clearly cited in your Methods section (and/or elsewhere in the manuscript) the following:

a) Circuit diagram (and preferably the printed circuit design), parts list (with costings, if possible), assembly instructions, 3D printing file (STL) for the case, and description of means of attachment, for the mobile node.

b) Code for the Raspberry Pi at the ground node, plus other details for constructing the ground node, including SD card, Wifi, housing (STLs?), etc.

c) In previous analogous instances, we have recommended that authors protect their hardware design using an Open Source Hardware, along the same lines as the CC-BY used for PLOS content: http://freedomdefined.org/OSHW - there are similar licences for software. We suggest that you do this.

d) Additional numerical data underlying the Figures (see Data Policy request below)

We expect to receive your revised manuscript within two weeks. Your revisions should address the specific points made by each reviewer. In addition to the remaining revisions and before we will be able to formally accept your manuscript and consider it "in press", we also need to ensure that your article conforms to our guidelines. A member of our team will be in touch shortly with a set of requests. As we can't proceed until these requirements are met, your swift response will help prevent delays to publication.

*Copyediting*

*Published Peer Review History*

*Early Version*

*Submitting Your Revision*

Sincerely,

Roli Roberts

Senior Editor

PLOS Biology

ETHICS STATEMENT:

The Ethics Statements in the submission form and Methods section of your manuscript should match verbatim. Please ensure that any changes are made to both versions.

-- Please include the full name of the IACUC/ethics committee that reviewed and approved the animal care and use protocol/permit/project license. Please also include an approval number.

-- Please include the specific national or international regulations/guidelines to which your animal care and use protocol adhered. Please note that institutional or accreditation organization guidelines (such as AAALAC) do not meet this requirement.

-- Please include information about the form of consent (written/oral) given for research involving human participants. All research involving human participants must have been approved by the authors' Institutional Review Board (IRB) or an equivalent committee, and all clinical investigation must have been conducted according to the principles expressed in the Declaration of Helsinki.

DATA POLICY:

We note that the raw tracking data and analysis code are provided in the GFBio depositions. However, we also require the numerical values that directly underlie your Figure to be made available in one of the following forms:

Regardless of the method selected, please ensure that you provide the individual numerical values that underlie the summary data displayed in the following figure panels as they are essential for readers to assess your analysis and to reproduce it: Figs 4 and 5 (I am assuming that Figs 2 and 3 can be plotted fairly directly from the GFBio tracking data). NOTE: the numerical data provided should include all replicates AND the way in which the plotted mean and errors were derived (it should not present only the mean/average values).

Please also ensure that figure legends in your manuscript include information on where the underlying data can be found (including GFBio URLs/DOIs), and ensure your supplemental data file/s has a legend.

---

## [Editor Report · Decision Letter 3]

24 Feb 2020

Dear Dr Ripperger,

On behalf of my colleagues and the Academic Editor, Graham K Taylor, I am pleased to inform you that we will be delighted to publish your Methods and Resources in PLOS Biology. 

Early Version

PRESS 

Kind regards,

Alice

Publication Assistant, 

PLOS Biology

on behalf of

Roland Roberts,

Senior Editor

PLOS Biology